# Environments, Behaviors, and Inequalities: Reflecting on the Impacts of the Influenza and Coronavirus Pandemics in the United States

**DOI:** 10.3390/ijerph17124484

**Published:** 2020-06-22

**Authors:** Jennifer D. Roberts, Shadi O. Tehrani

**Affiliations:** 1Department of Kinesiology, School of Public Health, University of Maryland, College Park, MD 20742, USA; 2School of Architecture and Environmental Design, Iran University of Science and Technology, Tehran 16846-13114, Iran; niusha.omidvar@yahoo.com

**Keywords:** influenza pandemic, coronavirus pandemic, health and social inequalities, social determinants of health

## Abstract

In the past century, dramatic shifts in demographics, globalization and urbanization have facilitated the rapid spread and transmission of infectious diseases across continents and countries. In a matter of weeks, the 2019 coronavirus pandemic devastated communities worldwide and reinforced the human perception of frailty and mortality. Even though the end of this pandemic story has yet to unfold, there is one parallel that is undeniable when a comparison is drawn between the 2019 coronavirus and the 1918 influenza pandemics. The public health response to disease outbreaks has remained nearly unchanged in the last 101 years. Furthermore, the role of environments and human behaviors on the effect and response to the coronavirus pandemic has brought to light many of the historic and contemporaneous inequalities and injustices that plague the United States. Through a reflection of these pandemic experiences, the American burden of disparity and disproportionality on morbidity, mortality and overall social determinants of health has been examined. Finally, a reimagination of a post-coronavirus existence has also been presented along with a discussion of possible solutions and considerations for moving forward to a new and better normal.

## 1. Introduction

### Pandemics: Environments, Behaviors, and Inequalities

“*An epidemic erodes social cohesiveness because the source of your danger is your fellow human beings, the source of your danger is your wife, children, parents and so on. So, if an epidemic goes on long enough, and the bodies start to pile up and nobody can dig graves fast enough to put the people into them, then morality does start to break down*.” [1]Dr. Alfred CrosbyAmerica’s Forgotten Pandemic: The Influenza of 1918

Across millennia, disease outbreaks have impacted societies and reinforced the human perception of frailty and mortality [2]. Dramatic demographic shifts in the past century have escalated the difficulty in containing a pandemic and the rise of globalization and urbanization have facilitated the rapid spread of pathogens from one continent to another in a flight of just a few hours. Yet, when a comparison is drawn between the 2019 coronavirus pandemic and the 1918 influenza pandemic, the most widely available tools to respond to these disease outbreaks have remained nearly the same. Public health interventions have been and are still the first line of defense against a pandemic in the absence of a vaccine [3]. Understanding that the environment plays a significant role in disease dynamics and in determining the health of individuals has always been an essential aspect to controlling and managing infectious diseases. For example, the history of urban planning in the past century has highlighted how built environments can have an effect on preventing both chronic and infectious diseases [4]. However, the function of human behaviors and social ideologies stemmed from the darker side of humanity must also be acknowledged as an important component to determining who lives and who survives during the deadliest periods of disease pandemics.

## 2. The 1918 Influenza Pandemic in the United States

### 2.1. Influenza Pandemic: An Epidemiological Overview

In the midst of World War I, the influenza pandemic spread worldwide during 1918-1919 and was outstanding in its lethality [5]. While there is not a universal consensus on the pandemic’s origin it is understood to have been was caused by a H1N1 subtype of an influenza A virus with avian and swine genetic origins [6,7,8]. Over the course of the pandemic, an estimated one third of the world’s population, approximately 500 million persons, became infected and the total deaths were estimated at 50 million or arguably as high as 100 million [7]. Historically, the curve of influenza deaths in humans has been U-shaped, with peaks among infants and the elderly. Nevertheless, this 1918 influenza pandemic exhibited a W-shape age-related fatality with the addition of a third peak among young adults [7,9]. As such, death rates for individuals 15 to 34 years of age were 20 times higher than in previous years [7,10].

The influenza pandemic occurred in three distinctive waves over a 12-month period in Europe, Asia and North America. The first wave of the pandemic, which began in spring 1918 and lasted approximately six months, was not particularly virulent and mimicked typical seasonal flu symptomology and mortality [7,8,11]. However, due to a possible viral mutation over the course of the summer and wartime troop movements, this first wave was then followed by a rapid succession of more fatal second and third waves in the autumn of 1918 (September–November) and the winter and spring of 1919 (January-May), respectively [7,12,13,14,15]. In the United States, the first outbreak of flu-like illnesses was detected in March 1918 with more than 100 cases at Camp Funston in Fort Riley, Kansas [16,17]. Within a week, the number of influenza cases quintupled throughout the army installation [16]. By the end of the month, approximately 1100 soldiers were hospitalized and 38 had died after developing pneumonia [18]. Nevertheless, thousands of soldiers were continually deployed for World War I and traveled across the Atlantic Ocean. In Europe, the disease first appeared in April 1918 among American soldiers in a military camp near Bordeaux, France [5,17]. The influenza presentation was in the form of a benign fever accompanied by cold-like symptoms and the attending American doctors believed the underlying cause was *Haemophilus influenzae*, a pathogenic bacterium formally called Pfeiffer’s bacillus [19]. The spring 1918 wave of influenza affected nearly 15,000 soldiers from April to July, but only five American soldiers died [19]. Although the mortality of this first pandemic wave was minor in comparison to the waves to come, Spain was the first European country in which the disease spread to wide sectors of the population, thus causing a significant level of mortality. In May and June 1918, there were 276 influenza deaths in Madrid, which represented a mortality rate of 0.42 deaths per 1000 people [19]. During this same period, another 852 unconfirmed influenza deaths were attributed to different respiratory system pathologies. If these unconfirmed deaths were combined with the confirmed influenza deaths, the mortality rate would have been 1.31 per 1000 people, the highest mortality figure traceable to this pandemic [19]. Up until this point, very little was written or documented about the strain of the outbreak or its epidemiologic indications due to wartime censorship, however this mortality rate from one single etiological agent could not be overlooked. In May 1918, the Spanish newspaper *El Sol* first published a report about a “sickness which ha[d] not yet been diagnosed by doctors” and headlines around the world labeled this as one of the worst pandemics in human history [19,20]. Thus, the derivation of “Spanish Influenza” was conceived [5].

In September 1918, the second, and most deadly, wave of the pandemic emerged within the United States [21]. Soldiers at Camp Devens, an army training camp just outside of Boston, Massachusetts, and sailors aboard the Receiving Ship at Commonwealth Pier in Boston began reporting flu-like symptoms [16,22]. By month’s end, more than 14,000 cases were reported at Camp Devens, approximately one-quarter of the total camp, and 757 soldiers succumbed to influenza [16]. The Massachusetts Department of Health informed local newspapers of the epidemic and stated that, “unless precautions are taken the disease in all probability will spread to the civilian population of the city” [22]. At the nearby Navy Radio School within Harvard University in Cambridge, Massachusetts, cases of influenza were reported among the group of 5000 young men studying radio communications. As 1918 concluded, Cambridge experienced 3014 cases of influenza among its meager population of 109,000 residents, ultimately leading to a mortality rate of 541 per 100,000 [23]. During this time in New York State, the Board of Health in New York City, added influenza to the list of reportable diseases, and required all cases to be isolated at home or in a city hospital [16]. Moreover, in a document entitled “*Fighting Influenza with Transit Systems*”, the Board of Health adopted a plan to set operation schedules for many business, which would then elongate the time window for transit service and reduce congestion [24,25]. In lieu of subway suspension, Royal Copeland, the City Health Commissioner “staggered workdays by industry, so that instead of everyone arriving at work at 9:00 [am], some were to come in later or earlier” [26]. Other cities, such as Seattle, Washington, followed suit in controlling the transmission of influenza within public transit environments by requiring passengers to wear face masks before boarding streetcars (Figure 1) [27].

Interestingly, Commissioner Copeland did not issue an order to close schools or the prohibition of public gatherings at theaters in New York City. He stated that his rational to not “[issue] general closing orders and making a public flurry over the situation was to keep down the danger of panic” and that he “wanted people to go about their business without constant fear and hysterical sense of calamity” [29]. Unlike New York City, public gatherings were prohibited and social distancing was encouraged with the closure of theaters, movie houses and night schools in other cities, namely Chicago, Illinois; Reno, Nevada; and Madison, Wisconsin [30,31]. There was also a call for volunteers to help nurse the infirmed in Chicago and other cities throughout the United States due to the severe deficit of professional nurses, a shortage resulting from the deployment of nurses to military camps and the failure to use trained African American nurses [16]. Despite the social distancing efforts employed in New York City and Chicago, Philadelphia, Pennsylvania choose to ignore these initiatives. On September 28, 1918, the city held its Fourth Liberty Loan Drive, a parade that gathered over 200,000 Philadelphians. Within days after the parade, over 1000 people were dead, another 200,000 cases of influenza were reported, every bed in the city’s 31 hospitals was filled, and the city was forced to close churches, schools, theaters and all other places of “public amusement” [22,32,33]. A week after the parade, 2600 Philadelphians had died [34]. With the pile up of corpses in Philadelphia, cold-storage plants were used as temporary morgues and a trolley car manufacturer donated 200 packing crates to be used as coffins [16]. By early October, single day influenza deaths reached record numbers in Philadelphia (289 deaths–October 6, 1918) and New York City (851 deaths–October 15, 1918) [22]. The virus continued to journey westward to Louisiana, Texas, New Mexico, Utah, California, and Washington. The wearing of six-ply gauze masks became mandatory in Washington and parts of California [16,22]. For example, in San Francisco, California, individuals were fined $5 U.S. dollars (USD), an equivalent of approximately $85 USD today, if they were seized in public without face masks [33]. In Salt Lake City, Utah, officials placed quarantine signs on the doors of 2000 homes where occupants had been infected with influenza [16]. The end of the second wave, which left its mark as being the deadliest of all three waves, coincided with the end of World War I on November 11, 1918 [11,21]. As people celebrated Armistice Day with parades and large gatherings, a public health calamity was lurking with an influenza resurgence in some cities [16,22]. By December 1918, the second wave of the pandemic had passed, but the health threat was far from over.

In Australia, a third wave erupted in January 1919 and eventually drifted back to Europe and the United States [18]. There were 1800 cases and 101 deaths reported in the first five days of January in San Francisco and on the other side of the country in New York City, 706 cases and 67 deaths were reported [16]. Even though eradication of influenza appeared in Louisiana and Washington, many residents in San Antonia, Texas began complaining that a new surge of influenza cases were not being reported [16,22]. As a precautionary measure, states like Massachusetts and Illinois began to request special appropriations for the study of influenza treatment and to train women as “practical nurses” in response to the nursing shortage [16]. The nursing shortage was so dire in some areas of the country that the American Red Cross asked local businesses to grant workers a day off if they volunteered in the hospitals at night [35]. The mortality rate of the third wave was just as high as the second wave, but the end of the war eliminated the conditions that enabled the rapid and widespread transmission of the virus. Deaths from the third wave paled in comparison to the annihilation during the second wave [18]. With one-quarter of the American population infected with influenza, the pandemic was affecting everyone. It was believed that President Woodrow Wilson contracted the virus in April 1919 while negotiating the Treaty of Versailles to end World War I [18]. The pandemic finally came to an end during the summer of 1919, although a very minor fourth wave appeared in spring 1920 throughout isolated areas of New York City. The pandemic ended simply because individuals who were infected either died or developed immunity. With an estimated death toll of 675,000 people in the United States, the pandemic lowered the average life expectancy by more than 12 years [10].

### 2.2. Influenza Pandemic: Environments and Behaviors

During World War I, the worst cases of the influenza pandemic came from overcrowded military barracks and ships with poor ventilation [36]. It was quickly found that one way to heal infected wounds was to expose patients to sunlight and open-air. In June 1915, Sir Arthur Everett Shipley, an eminent scientist of the time, determined that open-air treatment of infirmed and wounded soldiers at the First Eastern General Hospital was a success, particularly for those with influenza related pneumonia [37,38]. When the pandemic clinched the United States, health officials converted schools, halls, and large private houses to emergency and open-air hospitals [37]. As such, open-air hospitals became a common practice by the time of the 1918 pandemic [39,40]. A combination of outdoor air and sunlight reduced the number of infections and deaths reported at open-air hospitals. Specifically, medics found that regular meals, warmth, and plenty of fresh air and sunlight helped severely ill patients recover better than indoor nursed patients [41]. Researchers also revealed that healthcare facilities designed with high ceilings and large windows had the same effect as open-air hospitals [42]. Accordingly, natural ventilation and fresh air was the best way to prevent the transmission of influenza as well as other infectious diseases [43]. Open-air therapy was one of the popular techniques in dealing with common and often deadly respiratory infections of the time until antibiotics replaced it in the 1950s [44].

In the absence of a vaccine for novel pathogens and antibiotics to protect against influenza secondary bacterial infections, many authorities suggested non-pharmaceutical interventions for the control of a pandemic [45,46]. Social distancing had become the most well-established non-pharmaceutical public health measure to slow the spread of communicable diseases. Social distancing measures were implemented through the closing of schools, restaurants, and theaters as well as the banning and restriction of public gatherings and transportation among cities in the United States during the 1918 pandemic [3]. The point of initiation, duration and decision to relax social distancing measures profoundly impacted mortality rates during this period [47]. For example, cities such as Philadelphia (748 deaths per 100,000 at 24 weeks), that initiated social distancing measures later and for shorter periods tended to have spikes in deaths and overall higher death rates (Figure 2) [3,48]. In Philadelphia, the first cases of the influenza appeared on September 17, 1918 [47]. However, the Philadelphia Public Health Director, Dr. Wilmer Krusen, understated the risk of disease transmission. Dr. Krusen assured people that sick soldiers were suffering from casual seasonal influenza, which could be avoided by staying warm. He allowed large public gatherings, notably the Fourth Liberty Loan Drive parade on September 28, 1918 as mentioned previously. As a result, Philadelphia experienced a mortality acceleration that was 360% higher than the average American city [49]. In comparison, cities, such as Minneapolis, Minnesota (267 deaths per 100,000 at 24 weeks), that ordered social distancing measures sooner and for longer periods slowed the transmission rate and lowered overall death rates [3,48]. Furthermore, cities that relaxed social distancing measures prematurely, experienced multiple spikes in deaths. This was observed in St. Louis, Missouri. Due to its low death rate, the city lifted restrictions on public gatherings less than two months after the influenza outbreak began. However, a sharp and higher spike of new cases soon followed [3,48]. Research comparing 17 American cities has suggested that the implementation of early interventions to control the spread of influenza may have significantly decreased the rate of transmission and mortality observed [47]. Cities across the United States, including Milwaukee, Wisconsin and Indianapolis, Indiana, had the most effective interventions and reduced transmission rates by up to 30–50% [50]. The lower level of mortality was due to the quick public health response in the first days of the epidemic and the introduction of broad measures designed to promote social distancing [47,50].

Initially, the American medical community minimized the scale and scope of the influenza pandemic to create a false sense of security in the country’s largest cities, including Philadelphia [11]. Yet, this was an issue beyond the medical community; this was a public health crisis that brought to light urban environment deficiencies, namely the proliferation of municipal waste and unsanitary living conditions [51]. While not always well understood, the presence and threat of infectious diseases has shaped urban planning in the United States. During this early 20th century pandemic, urban environments in New York City, Chicago and other American cities were densely populated and characterized by crowded tenement housing in proximity to factories, animal yards, and slaughterhouses with little airflow or light [4,52]. Even before the 1918 influenza outbreak, cities were plagued with cholera, tuberculosis and typhoid epidemics [4]. Medical models, specifically the miasma theory, an idea that diseases were caused by a noxious form of “bad air”, prevailed, but there was also a sense that the congestion, pollution, lack of sunshine and poor airflow contributed to illness [4,52]. In response, urban planning efforts introduced zoning to spatially segregate residential, commercial and industrial uses and housing regulations to require light and airflow in order to improve overcrowding and unsanitary urban living conditions [4,52]. During the 1918 pandemic, urban reformers and engineers again tried to prevent infection by improving waste removal and housing quality and creating technological enhancements to control environmental nuisances and hazards, but trivial cooperation from municipal government leadership and interference of the private interest sector often blocked immediate and effective action [51]. As a result, the “schism in the public health community” ultimately impacted the lives of millions [53].

### 2.3. Influenza Pandemic: Societal Norms of Inequality

“Epidemic diseases [have not been] random events that afflicte[d] societies capriciously and without warning”; they typically unfold across a wide spectrum of communities that are diverse by race, age, gender, and socioeconomic status [54,55]. The 1918 pandemic affected the youngest and wealthiest people without any boundaries. Racially, the composition of the country during this time was predominately “black and white” and up until this point, medical and public health reports had documented that African Americans suffered higher morbidity and mortality rates for several diseases in comparison to White Americans [56]. Specifically, W.E.B. Du Bois, a prominent sociologist and civil rights activist, found through an analysis of the 1900 U.S. Census that the African American mortality rates were two to three times higher for several diseases, including tuberculosis, pneumonia, and diarrheal disease, in comparison to the White American mortality rates [57]. DuBois argued that these disparities reflected social conditions and not racial susceptibilities [57]. In poor and working-class urban neighborhoods, social conditions, like overcrowded housing, poor sanitation service, and unsafe drinking increased the vulnerability of influenza [51]. In 1910, the U.S. Census set 2500 as the minimum population threshold for the urban category and by the 1920 U.S. Census over 50% of the American population was defined as urban [58]. Before the onset of the pandemic, urban environments within American cities were composed of a predominantly (93.45%) White population [59]. During this period, more than 90% of the African American population lived in southern parts of the United States, but millions began migrating to the north in search of better job opportunities and living conditions [60]. With this First Great Migration from the rural south to northern cities, starting in 1910, a larger share of African Americans lived in urban areas during the 1918 pandemic [61]. Still, on average and in comparison to White Americans, fewer African Americans, less than 5%, lived in high population density urban areas, which were the epicenters of the outbreak [62]. Hence, the norms of disease morbidity and mortality for African Americans changed during the pandemic. During the second wave in fall 1918, African Americans had lower influenza morbidity and mortality rates, but a higher case fatality rate, in comparison to the White population [63]. This was an unexpected phenomenon as African Americans were expected to have higher influenza morbidity and mortality. Explanations for this 1918 pandemic norm crossover have included: (1) an unequal exposure based on where African Americans lived (e.g., urban vs. rural environments); (2) a higher exposure to the less virulent spring wave, which provided some African Americans with immunity; (3) a military segregation, which allowed for less crowding among African American soldiers; and (4) an underreporting of influenza cases and deaths for African Americans [63]. While there has been no evidence that the first spring wave was especially rampant in the south to induce a higher influenza exposure among the majority of African Americans living in rural southern areas, there has been a clear understanding that the First Great Migration left a large portion of African Americans in cramped, dilapidated, impoverished and segregated conditions with less access to sanitation and urban social programs in northern cities. The segregated African American neighborhoods may have functioned as a makeshift quarantine. Yet, in cities, such as Baltimore, Maryland and Detroit, Michigan, the segregation created crowded urban ghettoes where tuberculosis became one of the top three causes of death among urban African Americans [64]. Individuals with tuberculosis lung damage were more susceptible to influenza, which would explain why a higher proportion of African American influenza cases developed pneumonia and also had higher case fatality rates during the second wave [63]. The entrenched racial and economic discrimination within the United States seemed to have both protected and endangered African Americans to the 1918 pandemic.

Throughout World War I, many doctors had been called into military service and the shortage of the doctors, nurses, and medical staff caused hospitals to reject patients, particularly African Americans [56]. The American Red Cross served as the recruiter of nurses for the army. With the racial discriminatory practices of the American Red Cross, African American women who tried to gain entry as nurses were rebuffed. While the devastation of the pandemic forced the army to drop its ban on African American nurses, the racial discrimination still surpassed the need for healthcare in some areas. The November 2, 1918, edition of The Chicago Defender reported that the Dean of Ohio’s Hiram College refused to allow an African American nurse named Olive Walker to care for influenza patients at the college [56,65]. African American doctors also found the color line to be rigid in trying to care for White Americans during this time [66]. Racism and legalized segregation restricted African Americans and other non-White Americans access to healthcare professionals and facilities. In cities across the nation, African Americans received substandard care in segregated hospitals or in confined hospitals that served only African Americans. Many infected people died without seeing a doctor due to the sheer number of bodies that overwhelmed the capacity of hospitals and the heavy burden of the medical staff [67]. African American influenza patients exceeded the limit of the Frederick Douglass Memorial Hospital, one of Philadelphia’s two African American hospitals, and pushed the medical director to establish an “emergency annex” in an African American parochial school [56]. In Richmond, Virginia, African American patients received care in a hospital basement until the city opened a separate hospital staffed by African Americans at an elementary school [56,68]. The Baltimore Afro-American newspaper criticized Provident Hospital, the only African American facility in the city, for turning patients away [56,69]. A prohibition on public gatherings that included funerals and wakes prevented families from properly grieving. Many victims were buried without coffins or were thrown into unmarked burial grounds. Native Americans, who practiced different burial practices, became exposed to the virus at alarming rates. In September 1918, more than 2000 Navajos died of influenza in Apache County, New Mexico [22]. Across the country, mass graves were dug to accommodate the pile of corpses [36]. African American bodies, in particular, were neglected by the general public infrastructure. In Baltimore, sanitation department employees refused to dig graves for African American influenza victims after the city’s only African American cemetery could not accommodate any more graves [65].

The 1918 pandemic took a heavy human toll on entire families and communities. With the devastation and bereavement, violence erupted in some areas. People were shot for not wearing masks and rates of homicides and suicides skyrocketed [22]. White supremacist ideologies and racist scientific theories advanced bigoted and false notions of the biological, physiological, and moral inferiority of African Americans when they were blamed for the pandemic [56]. Throughout the First Great Migration period, African Americans were highly visible and thought of as “diseases”. The 1918 pandemic heightened these existing prejudices, which were further reinforced by contemporary newspaper headlines, such as “Rush of Negroes to City Starts Health Inquiry”; “Negros Arrive by Thousands—Peril to Health”; or “Negro Influx Brings Disease” [70]. The memory of the 1918 pandemic left a lasting blemish on public health planning and practice and reflected the moral relationships and deterioration of humanity during that period. The magnitude of events and death with the influenza pandemic have served as a reference point and hopefully as a severe, if not, “worst-case” scenario.

## 3. The 2019 Coronavirus Pandemic in the United States

### 3.1. Coronavirus Pandemic: An Epidemiological Overview

In December 2019, the novel coronavirus disease 2019 (COVID-19), caused by the severe acute respiratory syndrome coronavirus 2 (SARS-CoV-2), was detected in the Wuhan, Hubei Province of the People’s Republic of China [71]. In the following weeks, infections spread across China and other countries around the world [72,73]. COVID-19 was first confirmed to have spread to the United States in January 2020 when a 35-year-old man returned to Washington State on January 15, 2020 after traveling to visit family in Wuhan, China was diagnosed [74]. On January 30, 2020, the first case of person-to-person COVID-19 transmission was confirmed in the United States between a married couple after a 60-year-old wife returned from China [75]. She subsequently became symptomatic and transmitted the infection to her husband. By March 11, 2020, there were over 125,000 cases and 4500 deaths worldwide [76]. As such, the World Health Organization classified the COVID-19 outbreak a pandemic. As of June 21, 2020, there were over two million confirmed cases and 119,810 deaths in the United States [77]. These numbers, however, to do include the thousands of Americans, such as an estimated 5200 people in New York City, who died without being tested for COVID-19.

The coronavirus pandemic in the United States, which has the highest number of known cases in the world, exploded over the course of March and April 2020. At the start of March, most of the cases were tied to overseas travel and there was extremely limited testing available. In many parts of the country, such as in New York, Maryland, Indiana, North Carolina and Louisiana, there was a period when the number of cases was doubling every 7 to 14 days [77]. One precipitating source of community transmission unique to COVID-19 has been cruise ships. Cruise ships have previously been settings for infectious diseases outbreaks because of their compact closed environments, contact with travelers from many countries and crew transfers between ships. More than 800 confirmed COVID-19 cases occurred on the Diamond Princess and Grand Princess cruise ship voyages, the latter of which sailed roundtrip, departing San Francisco from February 11 to 7 March 2020 [78]. Four stops were made in Mexico (voyage A) and most of the 1,111 crew and 68 passengers from voyage A remained on board for a second voyage that departed San Francisco on 21 February 2020 (voyage B), with a planned return on 7 March 2020 [78]. A healthcare provider in California reported two patients, who had traveled on voyage A, with COVID-19 symptoms and one of whom with a confirmed positive COVID-19 test. Subsequently, more than 20 additional COVID-19 cases among voyage A passengers were confirmed in California. On 5 March 2020, a Centers for Disease Control and Prevention (CDC) response team was transported by helicopter to the Grand Princess cruise ship to collect specimens from 45 passengers and crew with COVID-19 respiratory symptoms. Among these 45 individuals, 21 (46.7%) had positive COVID-19 test results, which included two passengers and 19 crew members. This high attack rate among passengers and crew has been partially attributed to a high proportion of asymptomatic infections. Until disembarkation, passengers and symptomatic crew members were asked to self-quarantine in their cabins and room service replaced public dining. When the Grand Princess cruise ship docked in Oakland, California on 8 March 2020, everyone was transferred to land-based sites for a 14-day quarantine period. As of 21 March 2020, 78 (16.6%) of 469 persons with available test had positive COVID-19 test results. Additional COVID-19 cases linked to several additional cruises were also reported across the United States [78]. By 17 March 2020, the Cruise Lines International Association announced a 30-day voluntary suspension of cruise operations in the United States and the CDC issued a level 3 travel warning for all cruise travel to be deferred worldwide.

The first preliminary description of outcomes among COVID-19 patients in the United States was reported in March 2020 by the CDC. This report indicated that fatality was highest in persons aged ≥85, ranging from 10% to 27%. This was then followed by persons aged 65–84 years (3% to 11%), aged 55-64 years (1% to 3%), aged 20–54 years (<1%), and aged ≤19 years (0%) [79]. Pediatric (<18 years) COVID-19 cases, less than 2% of the total cases in the United States, have demonstrated much milder symptoms and may not even have a cough or fever [80]. Although severe COVID-19 illness leading to hospitalization, including intensive care unit (ICU) admission and death, can occur in adults of any age, 31% of cases, 45% of hospitalizations, 53% of ICU admissions, and 80% of deaths have occurred among adults aged ≥65 years [79]. Likewise, long-term residential care facilities were particularly at risk for COVID-19. In Washington, a female resident of a long-term care skilled nursing facility contracted COVID-19. An epidemiologic investigation determined that 129 cases of COVID-19 were associated with this facility, including 81 residents, 34 staff members, and 14 visitors. The majority of COVID-19 cases were women (65.1%) and the median age was 81 years (range = 54–100 years) among facility residents, 42.5 years (range = 22–79 years) among staff members, and 62.5 years (range = 52–88 years) among visitors. Out of the total 129 COVID-19 cases associated with this nursing facility, 56.8% of residents, 35.7% of visitors, and 5.9% of staff members were hospitalized. Ultimately, 23 persons (22 residents and 1 visitor) died as a result of COVID-19 [81]. While it is believed that limitations in effective infection control and prevention and staff members working in multiple facilities contributed to the spread within and between facilities, the existence of underlying medical conditions has been found to be a very important risk factor. Among these facility residents, the most common chronic underlying conditions were hypertension (69.1%), cardiac disease (56.8%), renal disease (43.2%), diabetes mellitus (37.0%), obesity (33.3%), and pulmonary disease (32.1%) [81].

By mid-March 2020, the entire United States, including four territories, had reported cases of COVID-19. On 7 April 2020, nationwide case doubling time was approximately 6.5 days, yet, for areas such as Louisiana the doubling time was 5.5. days [82]. Moreover, during this time Louisiana ranked 5th in the nation with approximately 16,200 confirmed cases and nearly 700 deaths, of which 70% were African American [77,83]. The explosion of cases in New Orleans has been anecdotally linked to the influx of travelers for Mardi Gras. Yet, this theory does not explain the situation of Shreveport, another Louisiana hotspot for COVID-19 cases [84]. Initially race and ethnicity data were not reported for COVID-19 cases, but then the COVID-19–Associated Hospitalization Surveillance Network (COVID-NET) was created using the existing infrastructure of the Influenza Hospitalization Surveillance Network (FluSurv-NET) and the Respiratory Syncytial Virus Hospitalization Surveillance Network (RSV-NET) in order to conduct population-based surveillance [85]. During the first month (1–30 March 2020) of surveillance in the United States, 1482 patients hospitalized with COVID-19 were surveilled. Among the COVID-19 patients with underlying conditions data, 89.3% had one or more underlying medical conditions. The most common conditions were hypertension (49.7%), obesity (48.3%), chronic lung disease (34.6%), diabetes mellitus (28.3%), and cardiovascular disease (27.8%) [85]. Hence, the majority of persons hospitalized with COVID-19 had underlying medical conditions. Among patients with race and ethnicity data, 45.0% were White, 33.1% were African American, 8.1% were Hispanic, 5.5% were Asian, and 7.9% were of other or unknown race [85]. Again, considering that African Americans only comprise 13% of the United States population, the overrepresentation of African American COVID-19 cases sheds light on issues that have often been overlooked.

### 3.2. Coronavirus Pandemic: Environments and Behaviors

Dr. Frank M. Snowden, a professor emeritus of history at Yale University stated in an interview with The New Yorker, that “[epidemics] reflect our relationships with the environment—the built environment that we create and the natural environment that responds” [86]. Communicable diseases have existed in humankind since the days of hunting and gathering, but when the shift to a more agrarian life occurred thousands of years ago, the formation of communities by way of built environments created ideal conditions for epidemics. Built environments, the man-made settings constructed and operated for the purpose of human activity, include the physical sites where humans live and work, and range in scale from homes in neighborhoods to buildings in cities. Some evidence has shown that high occupant density built environments can encourage the transmission of airborne pathogens that can cause the common cold, influenza, chicken pox, whooping cough, tuberculosis and even COVID-19 because of increased social interactions and direct contact between individuals [87]. Open interior spaces where there is a high degree of spatial connectivity and more opportunity for social encounters can facilitate the spread of these pathogens as well as others [88]. Therefore, the coronavirus pandemic has put a spotlight on the relationship between built environments, pathogenic infections, and public health.

On March 15, 2020, the CDC recommended social distancing, also called physical distancing in modern society [79,89,90]. Most states followed suit and issued stay-at-home orders with detailed directives such as “every person is ordered to stay at his or her place of residence except as necessary to perform essential activities” and essential activities included grocery shopping, obtaining medical prescriptions or service, providing care for others, or outdoor physical activity [91]. Many states, including Maryland and Massachusetts, prohibited gatherings of more than 10 people and mandated that individuals stay at least six feet apart [91]. States throughout the country had ordered most businesses to close, banned all non-essential gatherings of any size and implemented telework policies when possible. For instance, in New York, only pharmacies, some retailers, hospitals, manufacturing plants, and financial institutions were allowed to open [92]. The closure of the schools, businesses, restaurants without a carry-out or delivery options, and leisure activities, including theaters and fitness facilities, have in essence created an abandonment of many built environment structures. Many restaurants even with carry-out or delivery options have still needed to cut hours, switch to mobile payment systems and create new protocols of operation to face the challenges of the coronavirus pandemic [93]. Some retailers, namely grocery stores, have implemented a daily senior shopping hour for this at-risk population and added six feet apart points at checkout lines to help customers maintain a physical distance [94,95]. Amid the coronavirus pandemic, companies have enabled work-from-home structures to keep businesses running and assist employees with social distancing compliance. Nonetheless, working remotely has been a growing trend for a while and this pandemic essentially created millions of teleworkers overnight [96]. The public transportation infrastructure, another essential feature of built environments, has also been impacted by the severity of the coronavirus pandemic. In compliance with social distancing directives, many cities significantly reduced public transit service (e.g., subways, buses), thereby negatively affecting many individuals needing transit access. New York’s Metropolitan Transit Authority closed non-essential lines, concentrating resources to transit routes that supported essential personnel and those with urgent personal business [97]. New Orleans Regional Transit Authority has defined a set of essential destinations and eliminated routes that do not serve these locations [98]. The list of essential destinations included only healthcare facilities, pharmacies, and grocery stores. Many agencies, like the Chicago Transit Authority, have also imposed ad-hoc sanitation regimes to minimize the spread of the virus as well as the requirement of face masks for all riders [99]. The necessity of these preventative measures were underscored when a Detroit city bus driver vented on social media his anger over a coughing bus rider and then subsequently died 11 days later from COVID-19 [100]. In addition to a reduction of public transit in various municipalities throughout the United States, the service of commuter trains with Amtrak was also disrupted throughout the coronavirus pandemic due to social distancing and shelter-in-place orders. On 16 March 2020, Amtrak announced a 40% reduction, suspension or complete cancellation of various train lines throughout the northeast corridor of the country, including the Amtrak Hartford and Winter Park Express lines [101]. Amtrak has also required passengers to wear face masks [102]. Furthermore, the Department of Homeland Security (DHS) issued a Notice of Arrival Restrictions in early March 2020 whereby all airline passengers arriving from China, Iran, and certain European countries were required to travel through one of 13 airports where DHS had established enhanced entry screening capabilities [103]. By March 31, 2020 the Department of State disseminated a “Global Level 4 Health Advisory—Do Not Travel” and advised American citizens to avoid all international travel due to COVID-19 [104].

Airborne pathogens can be easily transmitted through indoor air and unfortunately indoor environments have not been designed to prevent the rapid dissemination of new diseases [105]. Ventilation and the pattern of airflow controls the direction of droplet distribution generated from the respiratory activities of building occupants. Thus, it is essential that environments with high occupant density and mobility, such as hospitals, schools, and offices, are designed to minimize the spread of airborne infections [105]. Administrators and building operators have been searching for ways to promote good design and amenity of built environments. Even before the coronavirus pandemic, new design and planning policies were trying to provide achievable guidance to environmental decision-makers, building operators and all indoor occupants in an attempt to minimize infectious disease transmission, particularly in dense urban environments [106]. Consequently, the new SARS-CoV-2 virus has made a profound impact on planning, environmental laws, and urban infrastructures. According to the University of Oregon’s Biology and the Built Environment Center, stringent social distancing measures and building cleaning efforts needed to be put in place to reduce exposure to SARS-CoV-2 [88]. Moreover, a more involved series of strategies, including changes to ventilation systems (e.g., increased outside air fractions to improve indoor air) and window operations (e.g., open windows to dilute indoor contaminants) have also emerged from the research center [88]. These modifications, as well as architectural techniques, combined with public health interventions can be used to efficiently disable diseases virulence and protect occupants in buildings from airborne pathogens [107,108].

Classifications and guidelines have shown that filtration as part of the ventilation system is a proven way to block the penetration of the outside pathogens into buildings [109,110]. Filtration can improve the indoor air quality and effectively reduce the pathogen loads released in the air [111,112]. Natural light and ultraviolet lights following proper protocols may also have the ability to inactivate SARS-CoV-2 [88]. Studies have shown that some viruses are susceptible to ultraviolet germicidal irradiation (UVGI). Factors, including humidity, airflow patterns, UVGI energy intensity and duration, have been able to impact UVGI effectiveness in inactivating pathogens [113,114,115,116]. Indoor humidity levels have also been discovered to be vital in the survivability of COVID-19. A research team found that indoor humidity levels of 40–60% may help minimize the spread and survival of SARS-CoV-2 and with little risk of indoor mold growth [88]. While, most designs of air-conditioning systems have not supported central humidification, a targeted in-room humidification may be necessary during episodes of pathogen transmission risk [88].

The coronavirus pandemic has exposed weaknesses in the American healthcare system, specifically the consortium of hospitals. There are more than 6000 hospitals with approximately 920,000 beds in the United States, but even before the pandemic there were fewer doctors and fewer beds per capita than most other developed countries [117,118]. According to data from the Organization for Economic Cooperation and Development, the United States has approximately 3 beds per 1000 people [117]. The higher rates of hospitalizations for chronic conditions that should not require hospitalization, including diabetes mellitus and asthma, has compounded the shortage of hospital beds during this pandemic. Furthermore, hospital emergency departments in many areas, including rural communities, have not been well equipped for modern ailments or the pressures of the coronavirus pandemic [119]. Designing guidance can aid healthcare facilities and clinics to cope with the increasing volume of COVID-19 cases and create small scale segregation rooms [120]. The guidance for controlling COVID-19 in healthcare settings indicated that patients should be isolated and placed in private rooms with a private bathroom. Moreover, airborne infection isolation rooms should be used for patients who will be undergoing aerosol generating procedures, such as intubation or chest compressions [121]. To combat the ongoing COVID-19 pandemic, a Dutch architecture firm suggested a studio prototype “vital house”, constructed quickly from sustainable prefabricated wooden materials. This “vital house”, a temporary healthcare center could provide ICUs, communication, and a large green patio to enable faster patient recovery [122]. Building pop-up triage tents immediately outside of hospitals has been a solution employed by medical centers throughout the Unites States. A 14-tent, 68-bed hospital in New York City’s Central Park and a 19-feet by 35-feet tent for 40 patients in Durham, North Carolina were constructed exclusively for COVID-19 patients when the number of cases needing care exceeded the hospital capacity [123,124]. These standing tents in empty spaces, under hospital overpasses and in proximity to emergency rooms have enabled healthcare workers to rapidly sequester crisis pandemic patients [125].

Another environmental issue that may have amplified COVID-19 transmission was the infection of healthcare workers who may have inadvertently spread the disease to patients. For example, COVID-19 was found to have spread across four buildings of Life Care Center, a consortium of skilled nursing facilities in Washington [126,127]. The course of events points to inadequate infection prevention and control measures in hospitals and other healthcare facilities. Stated by the World Health Organization, SARS-CoV-2 can survive on surfaces for a few hours up to several days. For example, SARS-CoV-2 RNA was identified on a variety of surfaces in cabins of both symptomatic and asymptomatic infected passengers up to 17 days after cabins were vacated on the Diamond Princess cruise ship, but before disinfection procedures had been conducted [78]. Reducing the number of flat surfaces where germs can sit has been deemed another way to keep pathogens and viruses from transferring between patients and healthcare workers around the building [125]. To control germs and bacteria on surfaces, designers have often avoided difficult-to-reach areas or tight corners in buildings and overly-complicated designs in high-touch surfaces [128]. According to Markovitz, hospital designers have attempted to find smart materials and infrastructure that can easily be disinfected and washed [125]. Also, certain materials already standardized in healthcare may find application in other public spaces, including antibacterial fabrics and finishes, non-porous materials that already exist (e.g., copper), as well as those that will inevitably be developed to facilitate cleaning and sanitizing [128]. Moving forward, it will also be important for architectures to immediately remove the high-touch design features of buildings [125]. Some large public spaces, such as airports, hotels, hospitals, gyms, and offices, have begun moving toward more automation to mitigate contagion. This coronavirus pandemic has revealed that using technology to remotely control the most-touched surfaces and developing touch-less technology (e.g., automatic doors; voice-activated elevators; cellphone-controlled hotel room entry; hands-free light switches and temperature controls) can significantly mitigate the spread of viruses [96].

### 3.3. Coronavirus Pandemic: Societal Norms of Inequality

Much of the United States, with the exception of essential workers, has undergone nearly three months of social distancing in addition to shelter-in-place orders during the coronavirus pandemic. Compared to others who are currently employed and working from home, essential workers must work outside the home. These workers, including healthcare workers, grocery store staff or transit operators, have been more likely to be African American and approximately 70% did not have a college degree [129]. Furthermore, one in four essential workers have said that they or someone in their family is a healthcare worker, an employment classification that has been considered essential throughout the United States [129]. Essential workers have been putting themselves at risk in order to keep others alive and maintain a functioning society through the country’s shelter-in-place orders. In addition to this risk, many essential workers have also been designated as economically vulnerable. One in three essential workers reported living in a household that made less than $40,000 USD a year and one in seven lacked health insurance [129]. Moreover, millions of essential workers rely on government assistance programs [130,131,132]. With a lack of options, many of these workers cannot afford to quit their jobs in spite of the heightened risk to SARS-CoV-2 exposure through their employment. A polarization has emerged among essential workers as a result of the inherent inequalities that persist within the United States. For example, there has been a clear economic and social distinction between a salaried pulmonologist who treats the respiratory system of COVID-19 patients and an hourly waged grocery store cashier who processes and bags customer purchases, yet both are considered essential. Aside from the difference in wages, the highly educated pulmonologist has been socially revered and showered with personal protective equipment campaigns while the grocery store cashier has been largely ignored, undervalued and unsupplied with simple face masks. It is clear that some essential positions require relatively few educational credentials or certifications and the skills necessary tend to be easy to learn and nontechnical. Therefore, the pool of these particular workers is large, and it has been easy for employers to hire and fire these workers at-will.

While simple economic concepts, such as supply and demand, can only partly account for the polarization of essential workers during the coronavirus pandemic, the norms of inequality within the United States contributes to the bulk of the circumstance. The jobs of essential workers have been more likely to be held by women, people of color, and immigrants. By the end of April 2019, the CDC reported that there were 4,913 essential workers diagnosed with COVID-19 in 115 meat or poultry processing facilities throughout 19 states and that 20 people had died [133]. It was also disclosed that these processing employees communicated through language and cultural barriers, lived in crowded, multigenerational housing, and commuted to work together. Likewise, workers were incentivized to work through illness thereby indicating a lack of appropriate healthcare options [133]. Many essential workers have been poorly compensated because they were members of marginalized and underrepresented groups who generally have less political and economic capital compared to the White American male. In addition to poor compensation, 13% of essential workers lack health insurance and over 30% of these workers would have to borrow money to pay for an unexpected $500 USD medical bill [129]. Within communities of color, many of whom are essential workers, as stated previously, COVID-19 testing inequities have been observed. In several areas throughout the country there has been a lack of access to COVID-19 walk-up testing as well as widespread testing [134]. In some observations, it has been reported that physicians may be less likely to refer African Americans for COVID-19 testing when they arrive with signs of infection [135,136,137]. Reasons for this have included the dismissiveness of African American health complaints and the inequitable race-based or income-based hierarchy of patients [138]. Pilot data have shown that, in comparison to White patients, African American patients with COVID-19 symptoms were six times less likely to receive testing or treatment [138]. Institutional and structural inequalities fueled by racism, sexism, and other forms of discrimination have impeded the ability of people of color to secure employment with livable wages, healthcare access and insurance, and an opportunity for upward mobility. Plus, given the societal expectation that these groups, and women in particular, provide service and care, this type of work has often been undervalued [139]. Women of color stand at the intersection of multiple oppressive identities and experience the amalgamated effects of racial, ethnic, and gender biases. Deep-rooted cultural attitudes and stereotypes about women of color have often devalued their work and deprioritized their needs [139]. This has been historically and contemporaneously shown through wage disparities. As an example, for every dollar earned by full-time, year-round working White men, Hispanic women earn 54 cents and African American women earn 62 cents while White women earn 79 cents [139,140]. The social distancing industry disruptions have greatly exacerbated the preexisting economic fragility of these women. As essential workers, many women of color have employment in healthcare or the service industry that places them on the frontlines of the coronavirus pandemic (Table 1) [139,141]. Although an overwhelming percentage of these women work as housekeeping cleaners (60%), nursing assistants (50%), and personal care aids (46%), women of color also disproportionately work in several industries that have also been hit hardest by pandemic related job losses [139,141]. Women of color (and all women) comprise 24% (54%) and 30% (80%) of the accommodations/food services and healthcare/social assistance industries, respectively, the two hardest hit industries of unemployment as reported by the U.S. Department of Labor [139,142].

The impact of social distancing is not just limited to economic and social inequalities; geographic disparities of COVID-19 cases and mortality have been just as alarming. Compared to other states, the first case of COVID-19 was not reported in Louisiana until March 9, 2020. Yet, as of June 21, 2020 with more than 49,497 confirmed cases and 3104 deaths, Louisiana ranked 7th in mortality rate (67 deaths per 100,000 people) [77,83,143]. These geographic variations do not necessary function exclusively as regional or spatial imbalances, but these variations are a manifestation of the residential segregation practices that have been and are still currently held in the United States [144]. Overwhelmingly, at one point 70% of Louisiana’s COVID-19 deaths were African American, despite this racial group comprising only 32% of the state’s population [77,83,145,146]. Although African Americans (54%) still maintain the highest percentage of COVID-19 deaths in Louisiana, the current racial breakdown of deaths are 44% (White (62% of Louisiana population)), 2% (Latinx (5% of Louisiana population)), and 0.8% (Asian (1.6% of Louisiana population)) [147]. Since the majority of deaths in Louisiana are African Americans, the intersection of race or racism with this COVID-19 epidemiological trend seems to be a strongly suggestive hypothesis of inequality especially since similar trends have been identified throughout the United States, including Illinois, Michigan, North Carolina, Wisconsin and even in the nation’s capital, Washington, DC [148,149,150,151]. While the exact cause for this racial disproportionality in COVID-19 cases and mortality has yet to be uncovered, the effect of underlying racial health disparities, namely kidney disease, asthma, diabetes mellitus and obesity, that are also risk factors for COVID-19, needs to be closely examined [152,153,154,155]. The notion that economic and social disparities create and perpetuate health disparities cannot be ignored. As discussed previously, institutional and structural inequalities have played a significant role in the social determinants of health, or the conditions of birth, growth, living, learning, working, playing and aging, for African Americans. For example, several African Americans as well as other communities of color tend to live in food, transit or recreational deserts, more polluted and densely impoverished areas, and have been raising children in single-parent households [156,157,158,159,160,161]. These social determinants of health have contributed to poorer COVID-19 outcomes. Additionally, many African Americans reside in areas where people have been unable to effectively social distance, such as prisons and homeless shelters [162]. Finally, systemic racism as a barrier to healthcare treatment for African Americans was present well before the coronavirus pandemic. Research has shown that African Americans rate health-related information they receive from family members and churches more highly due to an overarching sense of medical establishment distrust stemming from historical injustices (e.g., Tuskegee Syphilis Experiment) among this population [163,164,165,166,167]. Unfortunately, this has impeded access to medical care and magnified existing health disparities.

The coronavirus pandemic has also endangered communities of color through other very recognizable routes of discrimination. In early April 2020 the CDC took at 180 degree turn and instructed the use of “cloth face coverings”, such as bandanas, without considering the implications of gang affiliation, violence or racial profiling [168,169]. This created a sense of anxiety among communities of color, which was not unfounded. Specifically, African American men have been racially profiled when wearing surgical masks in an effort to reduce their COVID-19 risk, or have expressed a level of psychological distress due to the fear of being racially profiled [170,171]. A released video showed two African American men with facemask being followed in a Walmart store by a police officer in Wood River, Illinois. Although this incident occurred in March 2020, prior to the CDC face covering recommendation, all individuals throughout the country were well within their rights to use self-protecting efforts to reduce their COVID-19 risk. African Americans have not been the only targeted community of color. There has been an increase of anti-Asian discrimination because of the origination of COVID-19 in Wuhan, China [172]. The Federal Bureau of Investigation reported that Asian Americans were experiencing increased hate crimes due to the coronavirus global outbreak [173]. In Midland, Texas, three Asian American family members, including a 2-year-old baby and 6-year-old child, were stabbed at a Sam’s Club store and the suspect indicated that he attacked the family because he thought the family was Chinese, and infecting people with the coronavirus [174]. Also, in New York City teenagers attacked a 51-year-old woman on a city bus, spewed “anti-Asian statements”, and accused her of causing the coronavirus while another 13-year-old allegedly kicked a 59-year-old man for the same racist motives in a separate incident [174]. Even in the midst of these remarkable times, hatred and bigotry have continued to reign in the United States.

## 4. Creating a Post-COVID-19 New Normal

### Reflecting and Reimagining a New Post Pandemic Normal

“Epidemics are a category of disease that seem to hold up the mirror to human beings as to who we really are. That is to say, they obviously have everything to do with our relationship to our mortality, to death, to our lives. They show the moral relationships that we have toward each other as people, and we’re seeing that today” [86]. While there are distinct similarities and differences when comparing the 1918 and 2019 pandemics, the historical accounts of the influenza pandemic and the current coronavirus pandemic have demonstrate that poverty, inequality, and social determinants of health create and support the conditions for disease transmission and the exacerbation of existing health conditions and disparities (Table 2) [175]. Interestingly, the public health response to both pandemics, albeit tempered by varying levels of political resistance, impacted human behavior through mandated social distancing or the use of face masks. However, unlike the influenza pandemic, widespread daily protests and demonstrations within at least 140 American cities and throughout the world completely eclipsed and overshadowed public health orders for social distancing and public gathering prohibition during the coronavirus pandemic [176,177]. These protests rallied against the systemic police brutality of African Americans, specifically the killing of George Floyd by Minneapolis police, and in support of the Black Lives Matter movement [176,177]. This unique moment in history has illustrated the convergence of two public health crises ((1) coronavirus pandemic; (2) African American genocide), both of which are impacted by the annihilation of institutional, structural, systematic and systemic racism within the United States. As a new normal for post COVID-19 is reimagined the fundamental dynamics and entrenched ideologies of “acceptable living” and “equality for all” need to be addressed and strongly reexamined in order for the country to move forward.

Social distancing and shelter-in-place orders have been in place throughout the country since approximately the middle of March 2020. This has bought the country some time to curb the spread of the virus, limit the number of COVID-19 cases and deaths, reduce the healthcare system strain, and allow a reflection for the preparation of the next phase [178]. Following the pandemic containment and mitigation phases, the third suppression phase, which allows for a slow emergence from the quarantine status, requires essential elements in order to prevent an ascent of COVID-19 infections [178]. The elements include measurable benchmarks with surveilling epidemiological trends (e.g., decline in COVID-19 deaths for 14 days), fortifying the healthcare system (e.g., sufficient personal protective equipment for all healthcare workers) and transforming public health infrastructure (e.g., demonstrated ability to convey social distancing recommendations that change behavior in most residents) [178,179]. Once these elements have been implemented, aspects of social distancing can be relaxed. Yet, the old normal of living still cannot return and most likely will not return. During the initial reopening of establishments, gatherings should still be capped at 10 people and restaurants will need to adopt social distancing seating to welcome patrons. However, it is recommended by governmental and public health officials that individuals over 60 years of age and those who are medically vulnerable continue to shelter-in-place until the final phase of reopening [180]. A staggered or partial reopening approach with new environmental safeguards, such as temperature and ventilation checks and mandatory handwashing/sanitizing at every entry and periodically throughout the day may be used for schools and businesses [179]. There also may need to be a quarantine of travelers from high prevalence COVID-19 areas. Governmental regulations may institute proof of immunity documentation or spending fourteen days in isolation before entering a new country for travelers. Beyond a successful initial reopening and after at least four weeks of no significant increase in new COVID-19 cases, the six feet social distancing measure may be paused, but there should still be a ban on gatherings which exceed 50 people until four months after the initial reopening [179]. Throughout these phase transitions, it is critical for all public health leaders, political administrators, and the overall general public to work together in order to control, mitigate and eradicate COVID-19. As stated by Secretary-General António Guterres of the United Nations, the coronavirus pandemic is a public health emergency that became an economic, social and human rights crisis. Specifically, by working together and “respecting human rights in this time of crisis more effective and inclusive solutions for the emergency of today and the recovery for tomorrow” can be achieved [181].

Even with a new kind of normalcy, everyone may still practice consistent and better hygiene with handwashing, coughing, and sneezing, and greet and interact with each other differently [178]. People may need to create new ways of meeting with friends and colleagues. Impressively, within the past couple of decades, the sharing economy has expanded and created environments with new components of how multiple people share the same spaces. These shared spaces, including co-work environments (e.g., WeWork), rooms in homes (e.g., Airbnb), cars (e.g., Uber), bikes (e.g., Capital Bikeshare), and other elements of the built environment, which can increase the potential and opportunities for environmentally mediated pathways of pathogen exposure, will need to be reexamined [182]. The coronavirus pandemic also may accelerate the development of risk-mitigating efforts designed to reduce human-to-human interactions, upgrade building ventilation and physical barriers, and adopt automation technologies such as robotics and artificial intelligence vision systems [183]. Given the severity of the pandemic and in compliance with social distancing directives, many American cities significantly reduced public transit service as mentioned previously. While agencies, like the Chicago Transit Authority, implemented sanitation regimes to minimize the spread of the COVID-19, many transit agencies may move to a new standard where face masks will always be required once anyone boards a train or bus [99]. While social distancing measures prompted organizations to embrace videoconferencing, virtual classrooms and telemedicine, these types of interactions may become more of the norm moving forward. With the fear of contagion, companies have shifted their protocols of operation. Prior to the coronavirus pandemic, many businesses were already abandoning cash payments and in the wake of the outbreak, many more may permanently switch to digital only payments.

Structural building modifications (e.g., voice-activated elevators) or environmental adjustments (e.g., telework capabilities) may be a much easier acclimatization for many Americans in the wake of a post COVID-19 existence. Yet, changing perceptions, attitudes and behaviors that have been deeply entrenched in the American way of living will be a much more challenging hill to climb. Issues surrounding injustices related to many social determinants of health are not new and this pandemic has just illuminated these issues. The overwhelmingly disproportion COVID-19 mortality rates of African Americans only highlights the need to address longstanding inequalities. Racism in the United States has created inequality in access to healthcare, housing, wealth, education, and employment, all of which are social determinants of health that can either promote or undermine one’s health and longevity. In order to move forward, racism needs to be addressed and there needs to be new and tailored approaches to reach those who have been disadvantaged as well as the circumstances that have been generated through racist ideologies, practices and politices. For instance, when Louisiana initially launched drive-through testing, an African American 90-year-old woman walked a mile in the heat to get tested [184]. This demonstrated that this COVID-19 testing site was not accessible to many individuals who do not have cars or who live in transit deserts or who cannot afford private transit. Upon learning this, public health officials targeted zip codes with the highest rates of COVID-19 reported deaths and deployed testing in those areas. Population-based testing and contact tracing during this pandemic has been vital in order to predict where to allocate resources, control the spread of the virus and implement evidence-based interventions to high-risk areas. However, evidence-based interventions are only achievable when there are accurate data and evidence. It was not until early April 2020, that the CDC began including race and ethnicity data in its Morbidity and Mortality Weekly Report [85]. Many states have yet to release demographic data on deaths. While SARS-CoV-2 may be exploiting inequalities in the United States, this virus, similar to past as well as future pathogens, does not discriminate by race or any other social construct that man has created. If COVID-19 is not contained in the most vulnerable populations, it will continue to spread to all communities and alter the future of human existence.
*“This pesky flu’s all over town! And white and black and rich and poor are all included in its tour.”* [185]Author UnknownProse Poem on the 1918 Influenza Pandemic

## 5. Conclusions

For thousands of years disease outbreaks have devastated societies and the 2019 coronavirus pandemic is no different. As densely populated and urban environmental conditions shaped the 1918 influenza pandemic, the experience of the current day coronavirus pandemic is quite similar. While the influenza and coronavirus pandemics occurred nearly a century apart, there is one similarity of these events that is undeniable; the public health response to disease outbreaks has remained virtually unchanged with social distancing and face covering efforts. Unfortunately, the role of human behaviors and social ideologies stemmed from the darker side of humanity on the effect and response to the coronavirus pandemic has also not progressed considerably. This review has revealed how the American burden of inequalities and injustices, as related to environments and human behavior, significantly impacts morbidity and mortality rates among the most vulnerable communities. As observed with the 1918 influenza pandemic, the present-day coronavirus pandemic has revealed and exposed the many inequalities that inhabit the United States. However, an opportunity has been presented which can allow for improvements, advancements, and an overall social uplift of our society.

## Figures and Tables

**Figure 1 ijerph-17-04484-f001:**
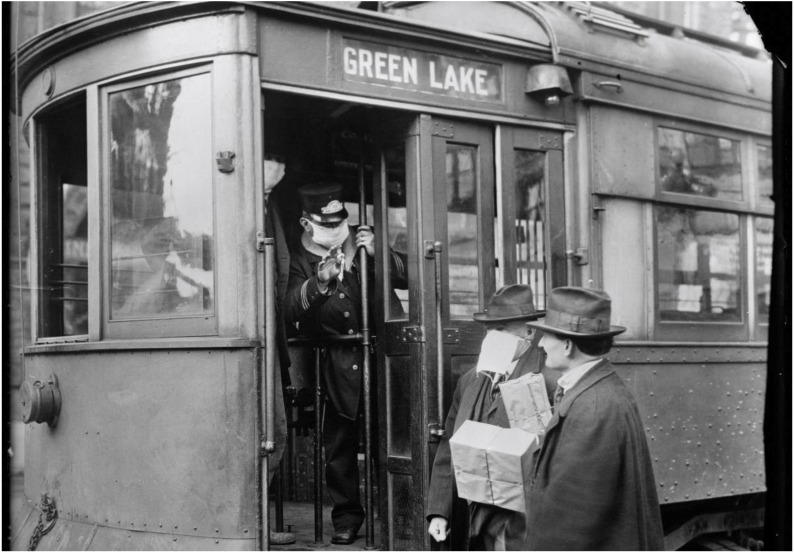
Seattle (Washington) streetcar conductor checks passenger face masks during the 1918 influenza pandemic (source: Library of Congress) [28].

**Figure 2 ijerph-17-04484-f002:**
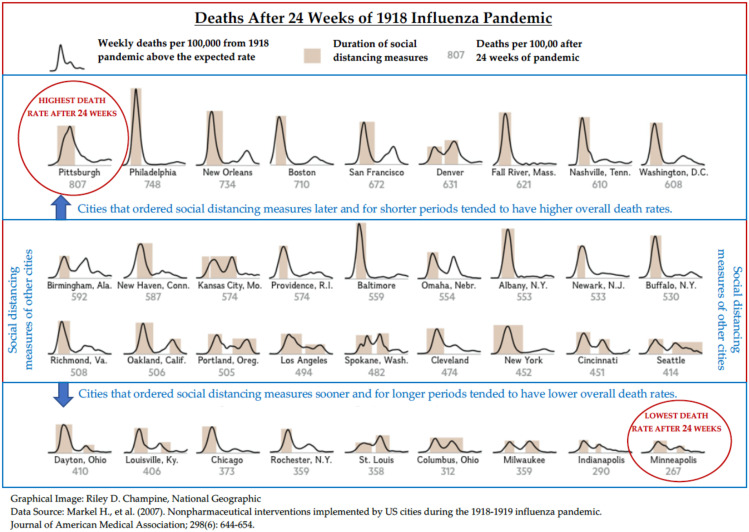
Death rates from 1918 influenza pandemic and duration of social distancing.

**Table 1 ijerph-17-04484-t001:** Top six occupations held by women of color in the United States [139,141].

	African American Women	Hispanic Women	Asian American Women	American Indian Women	Other Non-White Women	Two or More Races Women
**1**	nursing assistants	maids/housekeeping cleaners	registered nurses	cashiers	maids/housekeeping cleaners	cashiers
**2**	cashiers	cashiers	accountants/auditors	secretaries/administrative assistants	cashiers	registered nurses
**3**	customer service representatives	customer service representatives	cashiers	maids/housekeeping cleaners	nursing assistants	waitresses
**4**	registered nurses	secretaries/administrative assistants	manicurists/pedicurists	elementary/middle school teachers	registered nurses	elementary/middle school teachers
**5**	personal care aides	janitors/building cleaners	personal care aides	personal care aides	child care workers	customer service representatives
**6**	elementary/middle school teachers	retail salespeople	retail salespeople	registered nurses	retail salespeople	secretaries/administrative assistants

**Table 2 ijerph-17-04484-t002:** Comparison of the 1918 and 2019 pandemics in the United States.

Pandemic Characteristics	1918 Influenza Pandemic	2019 Coronavirus Pandemic
**Epidemiology**		
Disease Pathogen	Influenza A (H1N1)	SARS-CoV-2
Pandemic Origin	Fort Riley, Kansas, USA	Wuhan, China
Pandemic Waves	3	1 as of June 21, 2020
At Risk Population (Age)	15–34 years	65+ years
Life Expectancy Reduction	yes	no
Overall Mortality	675,000	119,810 as of June 21, 2020
**Environment**		
Hospital Environments	open-air	airborne infection isolation
Quarantine Measures	yes	yes
Military Environmental Risk	yes	no
Airplane Environment Risk	no	yes
Reduced Public Transit	yes	yes
Urban Environment Deficiencies	yes	yes
**Behavior**		
Social Distancing	yes	yes
Cloth Face Mask	yes	yes
Racial Profiling	yes	yes
Public Gatherings	no	no
Political Resistance	yes	yes
Widespread Protests	no	yes
**Inequalities**		
Healthcare Discrimination	yes (sanctioned)	yes
Health Provider Discrimination	yes (sanctioned)	yes
Economic Inequalities	yes	yes
Rural vs. Urban Inequalities	yes	yes
Preexisting Health Risks	yes	yes
Mortality Disparities (Race)	White Americans	African Americans

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
