# Peer review of "Environments, Behaviors, and Inequalities: Reflecting on the Impacts of the Influenza and Coronavirus Pandemics in the United States"

_ijerph, 2020, doi:10.3390/ijerph17124484_

Round 1
Reviewer 1 Report
In their manuscript, Roberts and Tehrani present a comparative Commentary on the 1918 influenza pandemic in the United States and COVID-19 in terms of built environments, behaviors, and issues of race and other social constructs. The manuscript is well written and extremely interesting. Overall, a very solid piece; however, some parts of the manuscript can be edited or updated, and a full comparison is needed from all the data between 1918 and 2019.
- Figure 1 – please ensure there is suffienct documentation of permissions to reproduce this figure in the current paper
- While it is understandable that pandemics move extremely quickly and data is hard to keep up in publication at this time of sanitary crisis, all efforts should be made to update the numbers presented in this section to reflect current knowledge.
- As the authors focus a good deal on closed environments, perhaps some anecdotes about transportation systems (airplanes, subways, buses, etc…) aside from cruise ships may be appropriate. This is touched up on lines 425-433, but no specific transmission of COVID (or other pathogens) are named with these systems.
- Lines 400-403: Perhaps this statement should be emphasized more as the main topic of the article and not just about COVID-19.
- Contrasts and similarities between the 1918 pandemic and the COVID response in the US need to be clearly laid out in an individual section (or, better yet, a table) for increased novelty of the findings from this extensive review and commentary.
Author Response
Dear Reviewer,
Thank you for your very helpful comments. I have addressed your comments and edits in the attached file.
Best,
Jennifer Roberts

Reviewer 2 Report
I start by congratulating the authors. This very good review paper outlines the evolution of the
different historical pandemics in the country.
The historical presentation of pandemics in the United States of America clearly illustrates the
weight of social and economics differences essentially in minority groups.
The scientific evidence analyzed and discussed in this review, allows a good understanding of the
association and relationship of the different actors in the field of public health, however, I make
only one suggestion:
From line 537 to 561, an overview of the type of access to health that the most vulnerable groups
around the present pandemic could be included.
It will be interesting to analyze from the perspective of the structure and functioning of the U.S.
health system, the social response and the effectiveness of the system itself. In addition to
justifying the authors' point of view, it will allow this approach to enrich the analysis of the
environment.
Finally, it is meritorious to read in the article the suggestions so diverse and complementary for
the return to the new normality after the confinement that will continue after the current
contingency and social asylum, however, it must be emphasized that the participation of the
different actors involved in the mechanisms of control, mitigation and eradication of infectious
health problems such as the one we are experiencing must be counted on.
Author Response

(The authors gave the same response as above.)

Reviewer 3 Report
Overall, the review offers a general public health and environmental perspective on the influenza and coronavirus pandemics with a subplot on conditions affecting African Americans. There is much good content, but the focus seems uneven. If the intent is to emphasize the African American experience, the review would be stronger with a more explicit and in-depth focus on the pandemic impact on African Americans.
Specific comments
Careful editing is needed to improve the manuscript in many places. Examples below .
- Dramatic language distracts and diminishes the message in several instances--suggest rewording:
- Line 640: “revolutionizing”
- Line 681: “wildly”
- Line 701 and repeated line 720: “will continue to spread and permanently alter human existence”
- Line 716: “drastically”
- Figure 1 needs better legends or titles to explain rows 2 and 3
- Check wording:
- Line 88 :used of double negative
- Line 149: meant ventilation instead of ventilator?
- Line 624: mention name of person whose quote is being used
- Line 625: demonstrated
- Line 632: “approximately nine weeks” as of what date?
- Line 637: sentence is unclear
- Line 642: implemented instead of “implanted”
- Line 646: state as recommendations by what authority, rather than “should”
- Line 655: may instead of “will”
Author Response

(The authors gave the same response as above.)

Reviewer 4 Report
This review article deals with some crucial events that took place during the influenza pandemic and the currently ongoing SARS-CoV2 pandemic, their impact on human health and behavior. Its a comprehensive review and excellently written to cover all old and recent literature. I have no major criticism about this article but do have following minor comments to improve the article:
Minor comments:
- Line 54 needs to be rewritten.
- There are formatting errors at few places in manuscript. These must be rectified.
- Lines 532-533 may need to be reconsidered. While it is understood that both a pulmonologist and a grocery store cashier who processes and bags customer purchases, are considered essential, the PPE kits cannot be provided to each and every person. This needs some rephrasing as a doctor closely interacts with patients who definitely reports with some symptoms of disease (here symptoms of viral infection) while a person at a grocery store might not be closely interacting with infected people. There will always be a major difference in the healthy to diseased ratio that these two category of essential personal deal everyday. The authors however, may keep their statement if they feel correct.
Author Response

(The authors gave the same response as above.)
